# Latent Domain Transfer: Crossing modalities with Bridging Autoencoders

## Abstract

Domain transfer is an exciting and challenging branch of machine learning because models must learn to smoothly transfer between domains, preserving local variations and capturing many aspects of variation without labels. However, most successful applications to date require the two domains to be closely related (e.g., image-to-image, video-video), utilizing similar or shared networks to transform domain-specific properties like texture, coloring, and line shapes. Here, we demonstrate that it is possible to transfer across modalities (e.g., image-to-audio) by first abstracting the data with latent generative models and then learning transformations between latent spaces. We find that a simple variational autoencoder is able to learn a shared latent space to bridge between two generative models in an unsupervised fashion, and even between different types of models (e.g., variational autoencoder and a generative adversarial network). We can further impose desired semantic alignment of attributes with a linear classifier in the shared latent space. Through qualitative and quantitative evaluations, we demonstrate that the proposed model preserves both locality and semantic alignment through the transfer process. Finally, the hierarchical structure decouples the cost of training the base generative models and semantic alignments, enabling computationally efficient and data efficient retraining of personalized mapping functions.

## 1 Introduction

Domain transfer has long captured the imagination of inventors and artists alike. The early precursor of the phonograph, the phonautograph, was actually inspired by the idea of "words which write themselves", where the shape of audio waveforms would transform into the shape of writing, capturing the content and character of the speaker's voice in shape and stroke of the written characters (Feaster, 2009). While perhaps fanciful at the time, modern deep learning techniques have shown similar complex transformations are indeed possible.

Deep learning enables domain transfer by learning a smooth mapping between two domains such that the variations in one domain are reflected in the other. This has been demonstrated to great effect within a data modality, for example transferring between two different styles of image (Isola et al., 2016; Zhu et al., 2017; Li et al., 2018; Li, 2018), video (Wang et al., 2018) and music (Mor et al., 2018). The works have been the basis of interesting creative tools, as small intuitive changes in the source domain are reflected by small intuitive changes in the target domain. Furthermore, the strong conditioning signal of the source domain makes learning transformations easier than learning a full generative model in each domain.

Despite these successes, this line of work in domain transfer has several limitations. The first limitation is that it requires that two domains should be closely related (e.g. image-to-image or video-to-video). This allows the model to focus on transferring local properties like texture and coloring instead of high-level semantics. For example, directly applying these image-to-image transfer such as CycleGAN (Zhu et al., 2017) or its variants to images from distant domains leads to distorted and unrealistic results (Li et al., 2018). This agrees with the findings of Chu et al. (2017) who show that CycleGAN transformations are more akin to adversarial examples than style transfer, as the model

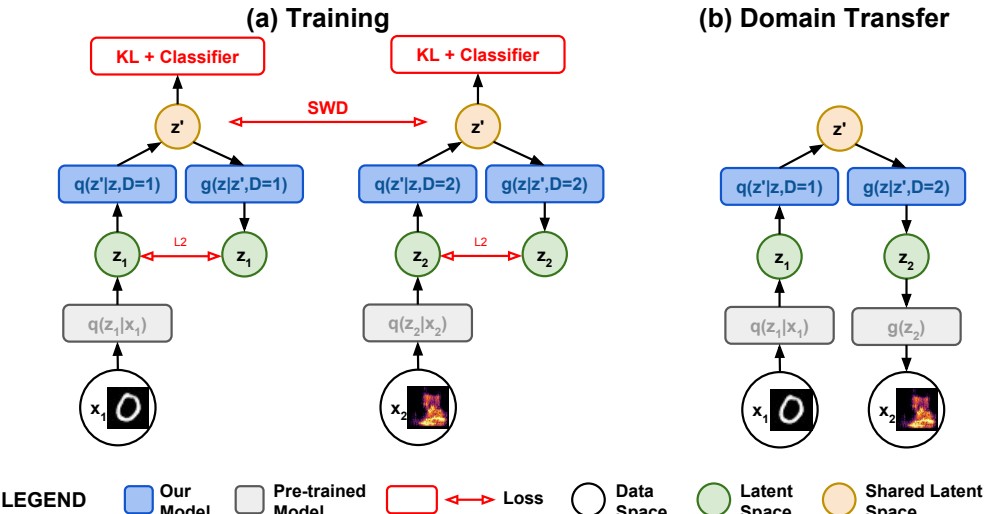

Figure 1: Architecture and training. Our method aims at transfer from one domain to another domain such that the correct semantics (e.g., label) is maintained across domains and local changes in the source domain should be reflected in the target domain. To achieve this, we train a model to transfer between the latent spaces of pre-trained generative models on source and target domains. **(a)** The training is done with three types of loss functions: (1) The VAE ELBO losses to encourage modeling of $z_1$ and $z_2$, which are denoted as L2 and KL in the figure. (2) The Sliced Wasserstein Distance loss to encourage cross-domain overlapping in the shared latent space, which is denoted as SWD. (3) The classification loss to encourage intra-class overlap in the shared latent space, which is denoted as Classifier. The training is semi-supervised, since (1) and (2) requires no supervision (classes) while only (3) needs such information. **(b)** To transfer data from one domain $x_1$ (an image of digit "0") to another domain $x_2$ (an audio of human saying "zero", shown in form of spectrum in the example), we first encode $x_1$ to $z_1 \sim q(z_1|x_1)$, which we then further encode to a shared latent vector $z'$ using our conditional encoder, $z' \sim q(z'|z_1, D = 1)$, where $D$ donates the operating domain. We then decode to the latent space of the target domain $z_2 = g(z|z', D = 2)$ using our conditional decoder, which finally is used to generate the transferred audio $x_2 = g(x_2|z_2)$.

learns to hide information about the source domain in near-imperceptible high-frequency variations of the target domain.

The second limitation is data efficiency. Most conditional GAN techniques, such as Pix2Pix (Isola et al., 2016) and vid2vid (Wang et al., 2018), require very dense supervision from large volumes of paired data. This is usually accomplished by extracting features, such as edges or a segmentation map, and then training the conditional GAN to learn the inverse mapping back to pixels. For many more interesting transformations, no such easy alignment procedure exists, and paired data is scarce. We demonstrate the limitation of existing approaches in Appendix C.

For multi-modal domain transfer, we seek to train a model capable of transferring instances from a source domain ($x_1$) to a target domain ($x_2$), such that local variations in source domain are transferred to local variations in the target domain. We refer to this property as *locality*. Thus, local interpolation in the source domain would ideally be similar to local interpolation in target domain when transferred.

There are many possible ways that two domains could align such that they maintain locality, with many different alignments of semantic attributes. For instance, for a limited dataset, there is no a priori reason that images of the digit "0" and spoken utterances of the digit "0" would align with each other. Or more abstractly, there may be no agreed common semantics for images of landscapes and passages of music, and it is at the liberty of the user to define such connections based on their own intent. Our goal in modeling is to respect the user's intent and make sure that the correct semantics (e.g., labels) are shared between the two domains after transfer. We refer to this property as *semantic alignment*. A user can thus sort a set of data points from in each domain into common bins, which we can use to constrain the cross-domain alignment. We can quantitatively measure the degree of

semantic alignment by using a classifier to label transformed data and measuring the percentage of data points that fall into the same bin for the source and target domain. Our goal can thus be stated as learning transformations that preserve locality and semantic alignment, while requiring as few labels from a user as possible.

To achieve this goal and tackle prior limitations, we propose to abstract the domain domains with independent latent variable models, and then learn to transfer between the latent spaces of those models. Our main contributions include:

- We propose a shared "bridging" VAE to transfer between latent generative models. Locality and semantic alignment of transformations are encouraged by applying a sliced-wasserstein distance, and a classification loss respectively to the shared latent space.

- We demonstrate with qualitative and quantitative results that our proposed method enables transfer both within a modality (image-to-image) and between modalities (image-to-audio).

- Since we training a smaller secondary model in latent space, we find improvements in training efficiency, measured by both in terms of the amount of required labeled data and well training time.

## 2 METHOD

Figure 1 diagrams our hierarchical approach. We first independently pre-train separate generative models, either a VAE or GAN, for both the source and target domain. For VAEs, data is encoded from the data domain to the latent space through a learned encoding function $z \sim q(z|x)$, and decoded back to the data space with a decoder function $\hat{x} \sim g(x|z)$. For GANs, we choose latent samples from a spherical Gaussian prior $z \sim p(z)$ and then use rejection sampling to only select latent samples whose associated data $x = g(z)$ is classified with high confidence by an auxiliary classifier.

We then add the bridging conditional VAE with shared weights, tasked with modeling both latent spaces $z_1$, $z_2$. The VAE has a single shared latent space $z'$, that corresponds to both domains. Sharing the weights encourages the model to seek common structures between the latent domains, but we also find it helpful to condition both the encoder $q_{shared}(z'|z, D)$ and decoder $g_{shared}(z|z', D)$, with an additional one-hot domain label, $D$, to allow the model some flexibility to adapt to variations particular to each domain. While the low-level VAEs have a spherical Gaussian prior, we penalize the KL-Divergence to be less than 1, allowing the models to achieve better reconstructions and retain some structure of the original dataset for the bridging VAE to model. Full architectural and training details can be found in the Appendix.

The conditional bridging VAE objective consists of three types of loss loss terms:

1. **Evidence Lower Bound (ELBO)**. Standard value that is maximized for training a VAE. For each domain $d \in \{1, 2\}$,

$$\mathcal{L}_d^{\text{ELBO}} = -\mathbb{E}_{z' \sim Z'_d} \left[ \log \pi(z_d; g(z', D = d)] + \beta_{\text{KL}} D_{\text{KL}} \left( q(z'|z_d, D = d) \| p(z') \right) \right]$$

where the likelihood $\pi(z; g)$ is a spherical Gaussian $\mathcal{N}(z; g, \sigma^2 I)$, and $\sigma$ and $\beta_{\text{KL}}$ are hyperparmeters set to 1 and 0.1 respectively to encourage reconstruction accuracy.

2. **Sliced Wasserstein Distance (SWD)** (Bonneel et al., 2015). The distribution distance between mini-batches of samples from each domain in the shared latent space $(z'_1, z'_2)$.

$$\mathcal{L}^{\text{SWD}} = 1/|\Omega| \sum_{\omega \in \Omega} W_2^2 \left( \text{proj}(z'_1, \omega), \text{proj}(z'_2, \omega) \right)$$

where $\Omega$ is a set of random unit vectors, $\text{proj}(A, a)$ is the projection of $A$ on vector $a$, and $W_2^2(A, B)$ is the quadratic Wasserstein distance.

3. **Classification Loss (Cls)**. For each domain $d \in \{1, 2\}$, we enforce semantic alignment with attribute labels $y$ and a classification loss in the shared latent space:

$$\mathcal{L}_d^{\text{Cls}} = \mathbb{E}_{z' \in Z'_d} H(f(z'), y)$$

where $H$ is the cross entropy loss, $f(z')$ is a one-layer linear classifier.

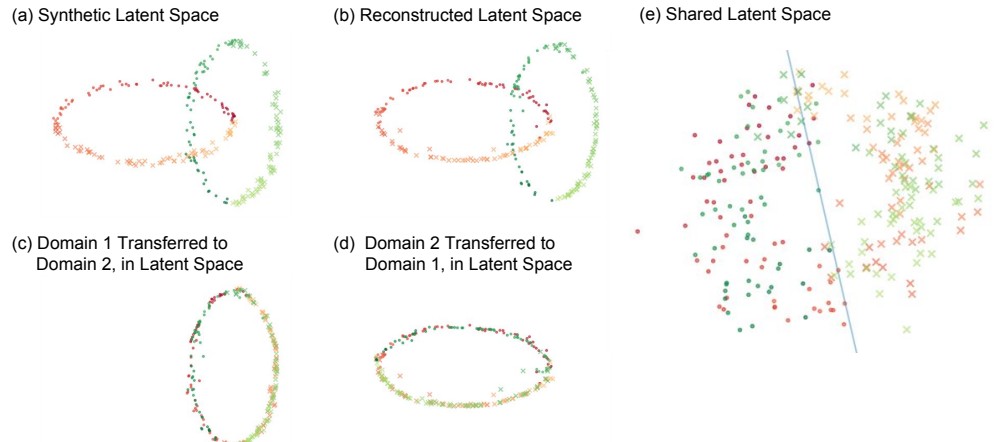

Figure 2: Synthetic data to demonstrate latent transformations. Both the low-level latent spaces and the shared latent space are two dimensional. Best viewed in color. **(a)** Synthetic data. The left red eclipse denotes the domain 1, while the right green eclipse denotes domain 2. The color gradient denotes the continuity of local changes. There are two classes, A (denoted as dots) and B (denoted as crosses), for both domains. Also note that for domain 1, label A and B are arranged up-and-down, while for domain 2 they are left-and-right. This is intentionally designed to force the model to learn a rotation instead of "cheating" by squeezing the shape of ellipses. **(b)** Reconstructions using shared VAE. **(c)** Domain 1 transferred to domain 2. Note that the transfer correctly handles classes as well as continuity of local changes **(d)** Domain 2 transferred to domain 1. Observation is similar to (c). **(e)** The shared latent space where the blue line is the decision boundary of the classifier. Here, the points from both domains are overlapping, class-separated, spread evenly, and maintain the continuity of color gradient.

Including terms for both domains, the total training loss is then

$$\mathcal{L} = (\mathcal{L}_1^{\text{ELBO}} + \mathcal{L}_2^{\text{ELBO}}) + \beta_{\text{SWD}}\mathcal{L}^{\text{SWD}} + \beta_{\text{Cls}}(\mathcal{L}_1^{\text{Cls}} + \mathcal{L}_2^{\text{Cls}})$$

Where $\beta_{\text{SWD}}$ and $\beta_{\text{Cls}}$ are scalar loss weights. The transfer procedure is illustrated Figure 2 using synthetic data. For reconstructions, data $x_1$ is passed through two encoders, $z_1 \sim q(z_1|x_1)$, $z' \sim q_{shared}(z'|z_1, D = 1)$, are reconstructed through two decoders, $\hat{z}_1 \sim g_{shared}(\hat{z}_1|z', D = 1)$, $\hat{x}_1 \sim g(\hat{x}_1|\hat{z}_1)$. For transformations, the encoding is the same, but decoding uses decoders (and conditioning) from the second domain, $\hat{z}_2 \sim g_{shared}(\hat{z}_2|z', D = 2)$, $\hat{x}_2 \sim g(\hat{x}_2|\hat{z}_2)$. Further analysis of this example and intuition behind loss terms is summarized in Figure 2 and detailed in Appendix A.

## 3 RELATED WORK

We discuss two aspects of existing research focuses that are related to our work, followed by how our work differentiates itself from them in order to deal with the challenge identified in this paper.

**Latent Generative Models:** Deep latent generative models are usually constructed to transfer a simple, tractable distribution $p(z)$ into the approximation of population distribution $p^*(x)$, through an expressive neural network function. Such models include VAE (Kingma & Welling, 2013) and GAN (Goodfellow et al., 2014). GANs are trained with an accompany classifier that attempts to distinguish between samples from the decoder and the true dataset. VAEs, in contrast, are trained with an encoder distribution $q(z|x)$ as an approximation to the posterior $p(z|x)$ using variational approximation through the use of evidence lower bound (ELBO). These classes of models have been thoroughly investigated in many applications and variants (Gulrajani et al., 2017; Li et al., 2017; Bikowski et al., 2018) including conditional generation (Mirza & Osindero, 2014), generation of one domain conditioned on another (Dai et al., 2017; Reed et al., 2016), generation of high-quality images (Karras et al., 2018) and long-range structure in music (Roberts et al., 2018). In terms of overall VAE structure, Zhao et al. (2017) studies options to build hierarchical VAEs.

**Domain Transfer:**   The domain transfer enables transfer between images (Isola et al., 2016; Zhu et al., 2017; Li et al., 2018; Li, 2018), audio (Mor et al., 2018) and video (Wang et al., 2018), intuitively mapping between two domains where the variations in one domain should be reflected in the other. Besides visually appealing results, domain transfer also enables application such as image colorization Zhang et al. (2017). Domain transfer is also proposed to be done through jointly training of generative models (Lu, 2018). Also, the behavior of domain transfer models also attracts attention. For example, Chu et al. (2017) suggests that image transfer does only local, texture level transfer.

To enable transfer between possibly drastically different domains, Our work proposes to use VAE in modeling the latent space of pre-trained generative models, in several aspects differentiating itself from related work. Generally, the modeling of latent space is different from modeling directly the data domains as most of latent generative models naturally do, also, the transfer in a more abstract semantics and between heterogeneous domains differs from most domain transfer method which focusing on locality and similar domains.

More specifically, regarding modeling latent spaces, Zhao et al. (2017) suggests training a hierarchical VAE on a single domain should be done end-to-end, whose extension to multiple domains seems non-trivial and likely to suffer from data efficient issues. Instead, our proposed work, though enabling separation of pre-trained model and conditional shared VAE, apply to domain transfer setting while overcoming this shortcoming. Moreover, regarding shared latent space, Liu et al. (2017) proposes to use shared latent space for two generative models on data domains. It requires joint training of generative models on both domains using dense supervision which is infeasible for drastically different domains. Our work that leverages pre-trained generative model and model the latent space instead of data domains addresses to this limitation.

## 4 EXPERIMENTS

### 4.1 DATASETS

While the end goal of our method is to enable creative mapping between datasets with arbitrary alignments, for quantitative studies we restrict ourselves to three domains where there exist a somewhat natural alignment to compare against:

1. **MNIST** (LeCun, 1998), which contains images of hand-written digits of 10 classes from "0" to "9".
2. **Fashion MNIST** (Xiao et al., 2017), which contains fashion related objects such as shoes, t-shirts, categorized into 10 classes. The structure of data and the size of images are identical to MNIST.
3. **SC09**, a subset of Speech Commands Dataset [1], which contains the record of audio of humans saying digits from "0" to "'9".

For MNIST and Fashion MNIST, we prepare VAE with MLP encoder and decoder following setting up in Engel et al. (2018). More specifically, we use stacks of fully-connected linear layers activated by ReLU, together with a "Gated Mixing Layer". The full network architecture of the bridging VAE is detailed in Appendix B. For SC09 we use the publicly available WaveGAN (Donahue et al., 2018)[2]. We would like to emphasize that we only use *class level* supervision for enforcing semantic alignment with the latent classifier.

We examine three scenarios of domain transfer:

1. MNIST ↔ MNIST. We first train two lower-level VAEs from different initial conditions. The bridging autoencoder is then tasked with transferring between latent spaces while maintaining the digit class from source to target.
2. MNIST ↔ Fashion MNIST. In this scenario, we specify a global one-to-one mapping between 10 digit classes and 10 fashion object classes (See Table 6 in Appendix for details).

---

[1]Available at `https://ai.googleblog.com/2017/08/launching-speech-commands-dataset.html`. License: `https://creativecommons.org/licenses/by/4.0/`

[2]Available at `https://github.com/chrisdonahue/wavegan`

The bridging autoencoder is tasked with preserving this mapping as it transfers between images of digits and clothing.

3. MNIST $\leftrightarrow$ SC09. For the speech dataset, we first train a GAN to generate audio waveforms (Donahue et al., 2018) of spoken digits. We chose to use a WaveGAN because we wanted a global latent variable for the full waveform (as opposed to a distributed latent code as in Engel et al. (2017)). It also gives us an opportunity to explore transferring between different classes of models. The bridging autoencoder is then tasked with transferring between a VAE of written digits and a GAN of spoken digits.

## 4.2 RECONSTRUCTION AND DOMAIN TRANSFER

For reconstructions and domain transfer, we present both qualitative and quantitative results. Quantitative measurements of semantic alignment are performed with pre-trained classifiers in each data domain. Given that the datasets have pre-aligned classes, when evaluating transferring from data $x_{d_1}$ to $x_{d_2}$, the reported accuracy is the portion of instances that $x_{d_1}$ and $x_{d_2}$ have the same predicted class.

Qualitative reconstruction results are shown in Figure 3 and the quantitative reconstruction accuracies are given in Table 1. For domain transfer, qualitative results are shown in Figure 4 and the quantitative transfer accuracies are given in Table 2. [3]

For both reconstruction and domain transfer, we see very good semantic alignment in the MNIST $\rightarrow$ MNIST and MNIST $\rightarrow$ Fashion MNIST, and reasonably good alignment in MNIST $\rightarrow$ SC09. The lower performance on SC09 is likely due to the comparatively poor performance of the base generative model, and would improve with a better audio generation model. For reference, the WaveGAN has to model a much larger signal than the MNIST VAEs (16000 dimensions vs. 768 dimensions).

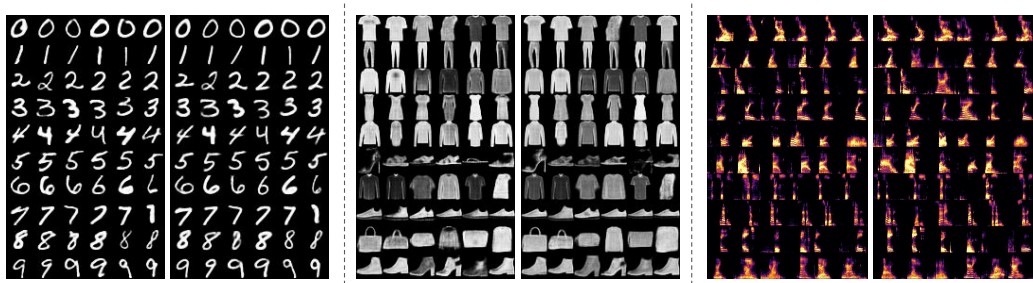

Figure 3: Qualitative Results for Reconstruction. Images are divided into three groups, representing MNIST, Fashion MNIST and SC09 reconstruction respectively from left to right. Specifically, for SC09 we show the log magnitude spectrum of the audio. Within each group, the left is the original and the right is the reconstruction. We see that the bridging autoencoder is able to archive high-quality reconstructions for MNIST and Fashion MNIST, and reasonable reconstructions for SC09.

| Data Domain | MNIST | Fashion MNIST | SC09 |
|:---:|:---:|:---:|:---:|
| Accuracy | 0.989 | 0.903 | 0.739 |

Table 1: Reconstruction Accuracy, for MNIST, Fashion MNIST and SC09 reconstruction respectively. The lower reconstruction accuracies of SC09 are due to the problem domain being much harder (16000 dimensions vs. 768 dimensions), and less well-modeled by the pre-trained GAN than the other domains are by the pre-trained VAEs.

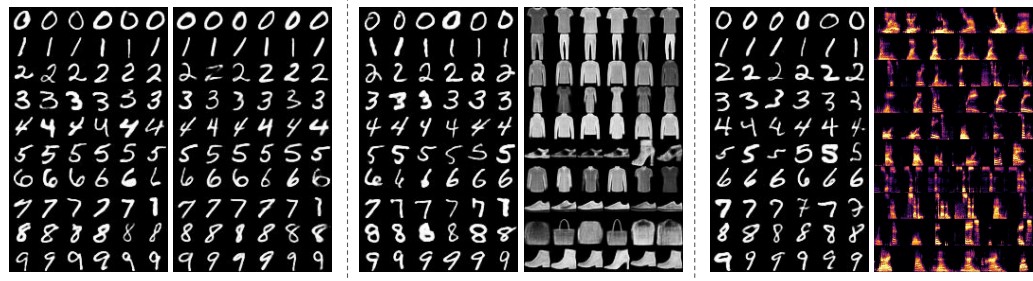

Figure 4: Qualitative Results for Domain Transfer. Images are divided into three groups, representing MNIST ↔ MNIST , MNIST ↔ Fashion MNIST and MNIST ↔ SC09 transfer respectively from left to right. Within each group, on the left is the data in the source domain and on the right is the data in the target domain. We see that transfer maintains the label, yet still maintains diversity of samples, reflecting the transfer of a broad range of attributes.

| Transfer | MNIST → MNIST | MNIST → Fashion MNIST | Fashion MNIST → MNIST | MNIST → SC09 | SC09 → MNIST |
|---|---|---|---|---|---|
| Pix2Pix(Isola et al., 2016) | - | 0.773 | 0.084 | × | × |
| CycleGAN (Zhu et al., 2017) | - | 0.075 | 0.132 | × | × |
| This work | 0.98 | **0.945** | **0.890** | 0.670 | 0.982 |

Table 2: Domain Transfer Accuracy for MNIST ↔ MNIST , MNIST ↔ Fashion MNIST and MNIST ↔ SC09 transfer respectively. We compare to pre-existing approaches trained on raw-pixels for MNIST ↔ Fashion MNIST only, MNIST → MNIST involves transferring between pretrained models with different initial conditions which is not directly comparable, and in MNIST → SC09, the two data domains were too distinct to provide any reasonable transfer with existing methods. Further comparisons can be found in Appendix C.

## 4.3 INTERPOLATION

Interpolation can act as a good proxy for locality and local smoothness of the latent transformations, as by definition good interpolations require that small changes in the source domain are reflected by small changes in the target domain. We show inter-class and inter-class interpolation in Figure 5 and Figure 6 respectively. Particularly, we are interested in two comparing three rows of interpolations: (1) the interpolation in the source domain's latent space, which acts a baseline for smoothness of interpolation for a pre-trained generative model, (2) transfer fixed points to the target domain's latent space and interpolate in that space, and (3) transfer all points of the source interpolation to the target domain's latent space, which shows how the transferring warps the latent space. We use spherical interpolation (e.g., $\sqrt{p}v_1 + \sqrt{(1-p)}v_2$)since we are interpolating in the Gaussian latent space.

Note in Figure 5 that the second and third rows have comparably smooth trajectories, reflecting that locality has been preserved. For inter-class interpolation in Figure 5 interpolation is smooth within a class, but between classes the second row blurs pixels to create blurry combinations of digits, while the full transformation in the third row makes sudden transitions between classes. This is expected from our training procedure as the bridging autoencoder is modeling the marginal posterior of each latent space, and thus always stays on the manifold of the actual data during interpolation.

## 4.4 DATA AND COMPUTATION EFFICIENCY, AND ABLATION ANALYSIS

Since our method is a semi-supervised method, we want to know how effectively our method leverages the labeled data. In Table 3 we show for the MNIST → MNIST setting the performance measured by transfer accuracy with respect to the number of labeled data points. Labels are distributed

---

[3]In Figure 3 and 4 we show the spectrum of audio samples for demonstration purpose. The corresponding audio samples themselves are available here: `https://drive.google.com/drive/u/8/folders/12u6fKvg0St6gjQ_c2bThX9B2KRJb7Cvk`

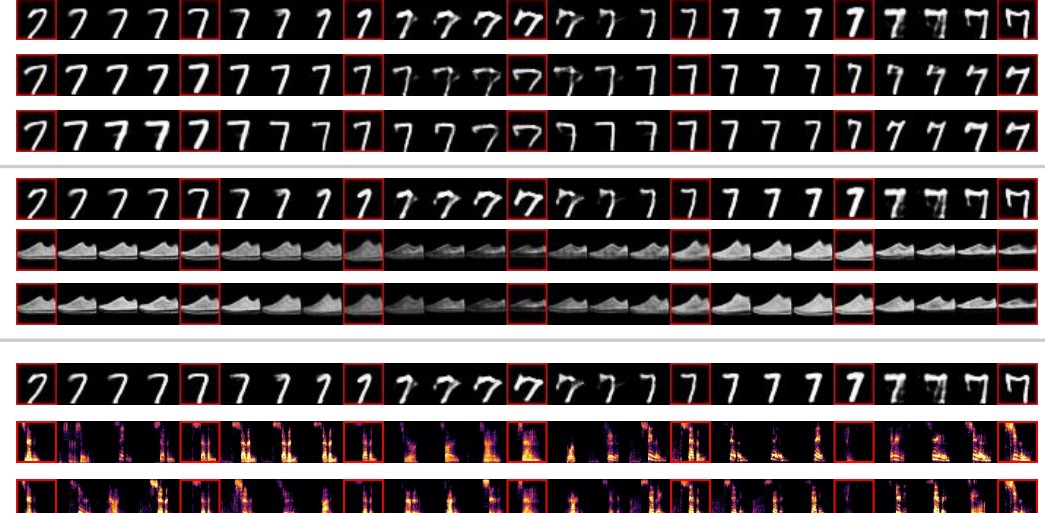

Figure 5: Intra-class Interpolation. Interpolations are divided into three groups, representing MNIST → MNIST , MNIST → Fashion MNIST and MNIST → SC09 transfer respectively from top to bottom. Images in a red square are fixed points in the interpolations, which means interpolation happens between two neighboring fixed points. In each group, there are three rows of interpolations: (1) Interpolate in source domain between fixed data points, (2) Transfer fixed data points in source domain to target domain and interpolate between transferred fixed points there, (3) Transfer all points in first row to the target domain. Note that in this intra-class setting, transferring preserves the smoothness of data when interpolating within one class.

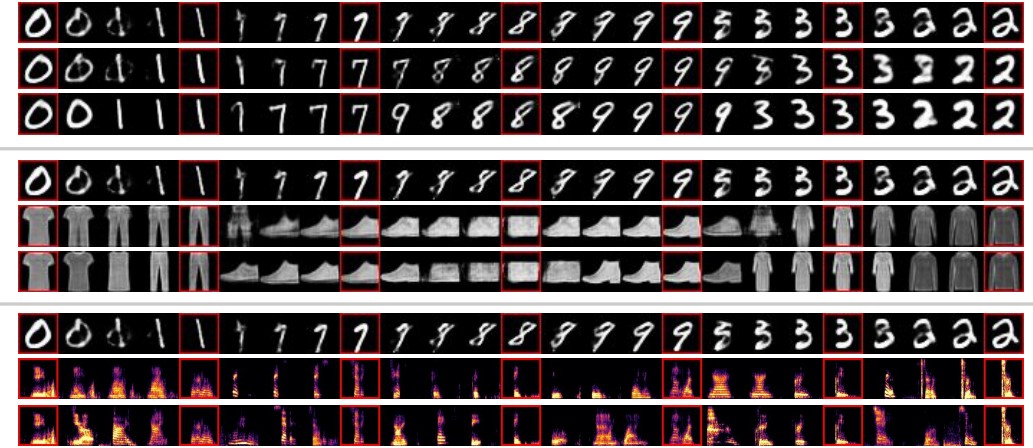

Figure 6: Inter-class Interpolation. The arrangement of images is the same as Figure 5, except that interpolation now happens between classes. It can be shown that, unlike regular generative model (row 1 and row 2 in each group) that exhibits pixel (data) level interpolation, especially the blurriness and distortion half way between instances of different labels, our proposed transfer (row 3) resorts to produce high-quality, in-domain data. This is an expected behavior since our proposed method learns to model the marginalized posterior of data distribution.

evenly among classes. The accuracy of transformations grows monotonically with the number of labels and reaches over 50% with as few as 10 labels per a class. Without labels, we also observe accuracies greater than chance due to unsupervised alignment introduced by the SWD penalty in the shared latent space.

| # Supervised | 0 | 10 | 100 | 1000 | 10000 | 60000 |
|---|---|---|---|---|---|---|
| Accuracy | 0.1390 | 0.339 | 0.524 | 0.6810 | 0.898 | 0.980 |

Table 3: MNIST → MNIST transfer accuracy as a function of labeled data points. The supervised data points are split evenly among all 10 classes.

Besides data efficiency, pre-training the base generative models has computational advantages. For large generative models that take weeks to train, it would be infeasible to retrain the entire model for each new cross-domain mapping. The bridging autoencoder avoids this cost by only retraining the latent transfer mappings. As an example from these experiments, training the bridging autoencoder for MNIST ↔ SC09 takes about one hour on a single GPU, while retraining the SC09 WaveGAN takes around four days.

Finally, we perform an ablation study to confirm the benefits of each architecture component to transfer accuracy. For consistency, we stick to the MNIST → MNIST setting with fully labeled data. In Table 4, we see that the largest contribution to performance is the giving the bridging VAE a domain conditioning signal, allowing it to share weights between domains, but also adapt to the specific structure of each domain. Further, the increased overlap in the shared latent space induced by the SWD penalty is reflected in the greater transfer accuracies.

| Data Domain | Vanilla, Unconditional VAE | Conditional VAE | Conditional VAE + SWD |
|---|---|---|---|
| Accuracy | 0.149 | 0.849 | 0.980 |

Table 4: Ablation study of MNIST → MNIST transfer accuracies.

## 5 CONCLUSION

We have demonstrated an approach to learn mappings between disparate domains by bridging the latent codes of each domain with a shared autoencoder. We find bridging VAEs are able to achieve high transfer accuracies, smoothly map interpolations between domains, and even connect different model types (VAEs and GANs). Here, we have restricted ourselves to datasets with intuitive class-level mappings for the purpose of quantitative comparisons, however, there are many interesting creative possibilities to apply these techniques between domains without a clear semantic alignment. As a semi-supervised technique, we have shown bridging autoencoders to require less supervised labels, making it more feasible to learn personalized cross-modal domain transfer based on the creative guidance of individual users.

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

## APPENDIX A    TRAINING TARGET DESIGN

We want to archive following three goals for the proposed VAE for latent spaces:

1. It should be able to model the latent space of both domains, including modeling local changes as well.

2. It should encode two latent spaces in a way to enable domain transferability. This means encoded $z_1$ and $z_2$ in the shared latent space should occupy overlapped spaces.

3. The transfer should be kept in the same class. That means, regardless of domain, $z$s for the same class should occupy overlapped spaces.

With these goals in mind, we propose to use an optimization target composing of three kinds of losses. In the following text for notational convenience, we denote approximated posterior $Z'_d \triangleq q(z'|z_d, D = d), z_d \sim q(z_d|x_d), x_d \sim p(x_d)$ for $d \in \{1, 2\}$, the process of sampling $z'_d$ from domain $d$.

**1. Modeling two latent spaces with local changes.**    VAEs are often used to model data with local changes in mind, usually demonstrated with smooth interpolation, and we believe this property also applies when modeling the latent space of data. Consider for each domain $d \in \{1, 2\}$, the VAE is fit to data to maximize the ELBO (Evidence Lower Bound)

$$\mathcal{L}_d^{\mathrm{ELBO}} = \mathbb{E}_{z' \sim Z'_d} \left[ \log \pi(z_d; g(z', D = d)) \right] - \beta_{\mathrm{KL}} D_{\mathrm{KL}} \left( q(z'|z_d, D = d) \| p(z') \right)$$

where both $q$ and $g$ are fit to maximize $\mathcal{L}_d^{\mathrm{ELBO}}$. Notably, the latent space $z$s are continuous, so we choose the likelihood $\pi(z; g)$ to be the product of $\mathcal{N}(z; g, \sigma^2 I)$, where we set $\sigma$ to be a constant that effectively sets $\log \pi(z; g) = ||z - g||_2$, which is the L2 loss in Figure 1 (a). Also, $D_{\mathrm{KL}}$ is denoted as KL loss in Figure 1 (a).

**2. Cross-domain overlapping in shared latent space.**    Formally, we propose to measure the cross-domain overlapping through the distance between following two distributions as a proxy: the distribution of $z'$ from source domain (e.g., $z'_1 \sim Z'_1$) and that from the target domain (e.g., $z'_2 \sim Z'_1$). We use Wasserstein Distance (Arjovsky et al., 2017) to measure the distance of two sets of samples (this notion straightforwardly applies to the mini-batch setting) $S'_1$ and $S'_2$, where $S'_1$ is sampled from the source domain $z'_1 \sim Z'_1$ and $S'_1$ from the target domain $z'_2 \sim Z'_d$. For computational efficiency and inspired by Deshpande et al. (2018), we use SWD, or Sliced Wasserstein Distance (Bonneel et al., 2015) between $S'_1$ and $S'_2$ as a loss term to encourage cross-domain overlapping in shared latent space. This means in practice we introduce the loss term

$$\mathcal{L}^{\mathrm{SWD}} = \frac{1}{|\Omega|} \sum_{\omega \in \Omega} W_2^2 \left( \mathrm{proj}(S'_1, \omega), \mathrm{proj}(S'_2, \omega) \right)$$

where $\Omega$ is a set of random unit vectors, $\mathrm{proj}(A, a)$ is the projection of $A$ on vector $a$, and $W_2^2(A, B)$ is the quadratic Wasserstein distance, which in the one-dimensional case can be easily solved by monotonically pairing points in $A$ and $B$, as proven in Deshpande et al. (2018).

**3. Intra-class overlapping in shared latent space.**    We want that regardless of domain, $z$s for the same class should occupy overlapped spaces, so that instance of a particular class should retain its label through the transferring. We therefore introduce the following loss term for both domain $d \in \{1, 2\}$

$$\mathcal{L}_d^{\mathrm{Cls}} = \mathbb{E}_{z' \in Z'_d} H(f(z'), l_{x'})$$

where H is the cross entropy loss, $f(z')$ is a one-layer linear classifier, and $l_{x'}$ is the one-hot representation of label of $x'$ where $x'$ is the data associated with $z'$. We intentionally make classifier $f$ as simple as possible in order to encourage more capacity in the VAE instead of the classifier. Notably, unlike previous two categories of losses that are unsupervised, this loss requires labels and is thus supervised.

In Figure 7 we show the intuition to design and the contribution to performance from each loss terms.

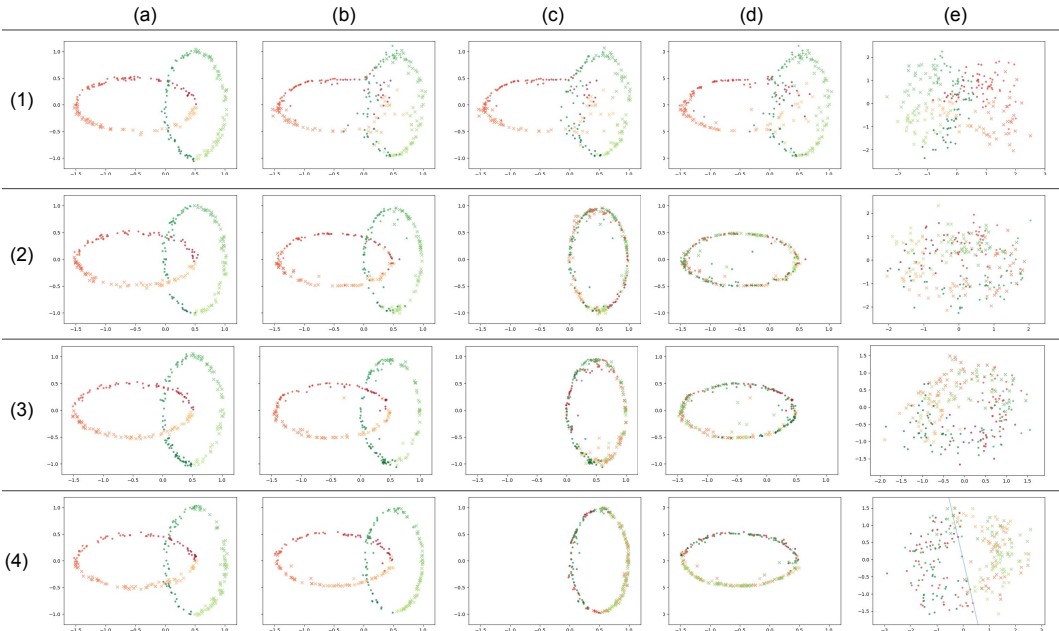

Figure 7: Synthetic data to demonstrate the transfer between 2-D latent spaces with 2-D shared latent space. Better viewed with color and magnifier. Columns (a) - (e) are synthetic data in latent space, reconstructed latent space points using VAE, domain 1 transferred to domain 2, domain 2 transferred to domain 1, shared latent space, respectively, follow the same arrangement as Figure 2. Each row represent a combination of our proposed components as follows: **(1)** Regular, unconditional VAE. Here transfer fails and the shared latent space are divided into region for two domains. **(2)** Conditional VAE. Here exists an overlapped shared latent space. However the shared latent space are not mixed well. **(3)** Conditional VAE + SWD. Here the shared latent space are well mixed, preserving the local changes across domain transfer. **(4)** Conditional + SWD + Classification. This is the best scenario that enables both domain transfer and class preservation as well as local changes. It is also highlighted in Figure 2. An overall observation is that each proposed component contributes positively to the performance in this synthetic data, which serves as a motivation for our decision to include all of them.

## APPENDIX B    MODEL ARCHITECTURE

The model architecture of our proposed VAE is illustrated in Figure B. The model relies on Gated Mixing Layers, or GML. We find empirically that GML improves performance by a large margin than linear layers, for which we hypothesize that this is because both the latent space ($z_1, z_2$) and the shared latent space $z'$ are Gaussian space, GML helps optimization by starting with a good initialization. We also explore other popular network components such as residual network and batch normalization, but find that they are not providing performance improvements. Also, the condition is fed to encoder and decoder as a 2-length one hot vector indicating one of two domains.

For all settings, we use the dimension of shared latent space 100, $\beta_{\mathrm{SWD}} = 1.0$ and $\beta_{\mathrm{CLs}} = 0.05$,

Specifically, for MNIST $\leftrightarrow$ MNIST and MNIST $\leftrightarrow$ Fashion MNIST, we use the dimension of shared latent space 8, 4 layers of FC (fully connected layers) of size 512 with ReLU, $\beta_{\mathrm{KL}} = 0.05$, $\beta_{\mathrm{SWD}} = 1.0$ and $\beta_{\mathrm{CLs}} = 0.05$; while for MNIST $\leftrightarrow$ SC09, we use the dimension of shared latent space 16, 8 layers of FC (fully connected layers) of size 1024 with ReLU $\beta_{\mathrm{KL}} = 0.01$, $\beta_{\mathrm{SWD}} = 3.0$ and $\beta_{\mathrm{CLs}} = 0.3$. The difference is due to that GAN does not provide posterior, so the latent space points estimated by the classifier is much harder to model.

For optimization, we use Adam optimizer(Kingma & Ba, 2014) with learning rate 0.001, $beta_1 = 0.9$ and $beta_2 = 0.999$. We train 50000 batches with batch size 128. We do not employ any other tricks for VAE training.

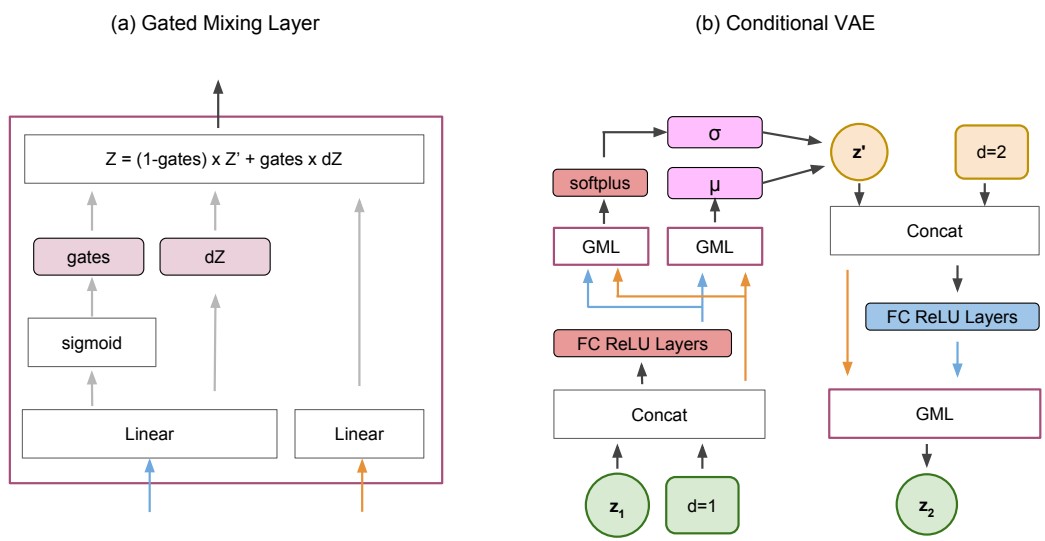

Figure 8: Model Architecture for our Conditional VAE. **(a)** Gated Mixing Layer, or GML, as an important building component. **(b)** Our conditional VAE with GML.

## APPENDIX C    COMPARISON TO EXISTING APPROACHES

We compare our results with two existing approaches, Pix2Pix (Isola et al., 2016) on the left and CycleGAN (Zhu et al., 2017) on the right, on the same MNIST $\leftrightarrow$ Fashion MNIST transfer settings used in Figured 4. We show qualitative results from applying Pix2Pix and CycleGAN in Figure 9, which can be compared with Figured 4, as well as quantitative results in Table 5. Both qualitative and quantitative results shows the limitation of existing methods and our proposed approach's advantage over them.

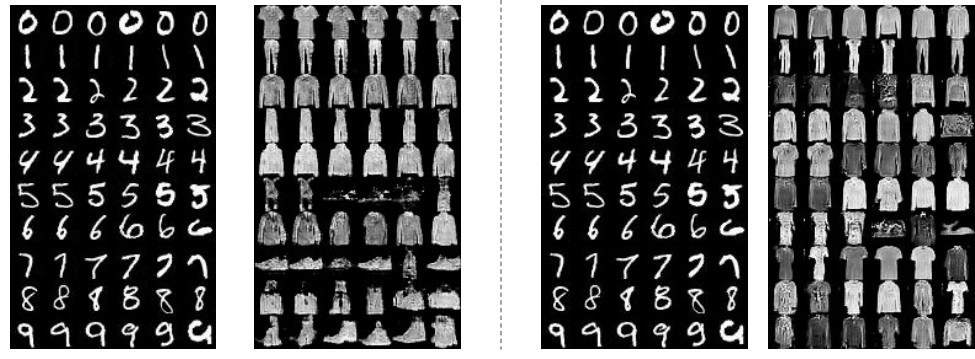

Figure 9: Qualitative results from applying Pix2Pix (Isola et al., 2016) on the left and Cycle-GAN (Zhu et al., 2017) on the right, on the same settings used in Figured 4. Visually, both existing transfer approaches suffer from less desirable overall visual quality and less diversity in local changes, compared to our proposed approach. Particularly, Pix2Pix more or less makes semantic labels correct but suffers from mode collapses in each label, while CycleGAN has slightly better quality but suffers from label collapse, which is observable here that most of digits are transferred to Dress and leads to bad transfer accuracy.

| Method | Transfer Accuarcy | FID (Fréchet Inception Distance) |
|---|---|---|
| Pix2Pix(Isola et al., 2016) | 0.773 | 0.0786 |
| CycleGAN (Zhu et al., 2017) | 0.075 | 0.3333 |
| This work | **0.945** | **0.0055** |

Table 5: Quantitative results of methods using Transfer Accuracy and Fréchet Inception Distance (FID). Transfer Accuracy is calculated using the same protocol as Table 2 where a higher value indicates the better performance, while FID is computed using the activation of the classifier on the target domain (Fashion MNIST) where a lower value indicates a better image quality. Quantitatively both existing methods perform worse than our proposed method. Here Pix2Pix more or less makes semantic labels correct but still suffers from lower accuracy and image quality. while CycleGAN suffers from label collapse, which leads to an even lower transfer accuracy and FID.

## APPENDIX D    SUPPLEMENTARY FIGURES

| MNIST Digits | Fashion MNIST Class |
|---|---|
| 0 | T-shirt/top |
| 1 | Trouser |
| 2 | Pullover |
| 3 | Dress |
| 4 | Coat |
| 5 | Sandal |
| 6 | Shirt |
| 7 | Sneaker |
| 8 | Bag |
| 9 | Ankle boot |

Table 6: MNIST Digits to Fashion MNIST class Mapping. This mapping is made according to Labels information available at `https://github.com/zalandoresearch/fashion-mnist`

