# OpenReview forum: "Latent Domain Transfer: Crossing modalities with Bridging Autoencoders"
_ICLR.cc/2019/Conference_

### Official Review · AnonReviewer1 · 2018-11-02
**poor organization, trivial techical implementation**

**Rating:** 4
**Confidence:** 4

**Review:**

In this paper, the authors have proposed a cross domain transferring methods, supervised by three category of losses. The experiments somewhat demonstrate the effective of this method. However, this paper still suffers from some drawbacks as below:
The paper is not well-organized, the structure of the paper need improving. For example, the related work is put almost at the end of the paper and the tables and figures are hard to follow sometimes.
The technical implementation of the proposition is somewhat trivial. Why the generative model should be pre-trained. Why not try in the end-to-end way.
The experiments are not convincing. The authors argue that CycleGAN suffers from some drawback. Why do not the authors compare with CycleGAN in this paper? By the way, the authors also need to compare with more state-of-the-art methods, such as StarGAN.
Some implementation details are not clearly stated. For example, the authors say “Our goal can thus be stated as learning transformations that preserve locality and semantic alignment, while requiring as few labels from a user as possible.” So, how many labeled samples are used in Table 2?

---

> ### Author Response · Authors · 2018-11-26
> **Response to AnonReviewer1**
>
> Thank you for your time and insight in your review. We've done our best to address your concerns with paper revisions and in the comments below:
>
> > “The paper is not well-organized, the structure of the paper need improving.”
>
> We agree with your assessment and thank you for your helpful suggestions. The updated draft has been extensively revised and restructured. For example, following your advice, we have moved the new related work section to follow the methods, and added more details to the figure and table captions to make their explanations self contained.
>
> > “The technical implementation of the proposition is somewhat trivial. Why the generative model should be pre-trained. Why not try in the end-to-end way. “
>
> We would like to highlight that the problem this paper addresses (cross-modal domain transfer) is difficult and, to the best of our knowledge, relatively unexamined in the literature. We believe it is actually a desirable feature, and not a fault, that the proposed method is fairly straightforward and easy to implement.
>
> The point about end-to-end training is well-taken. For simpler problems, like MNIST <-> MNIST, and MNIST<-> Fashion MNIST, end-to-end training is indeed tractable. However, we would like to highlight some advantages of the multi-step approach. First, this approach allows us to combine models that use dramatically different training procedures. We demonstrate that in this paper by transferring between a maximum-likelihood trained VAE and an adversarial-trained GAN. Second, for large generative models that take weeks to train, it would be infeasible to retrain the entire model for each new domain mapping. As a small example from this paper, training the bridging autoencoder from MNIST->SC09 takes ~1 hour on a single gpu, while retraining the SC09 WaveGAN takes ~4 days. We have also restricted ourselves to intuitive class-level mappings for the purpose of quantitative comparisons in this paper, but in a creative application it is likely each user would prefer their own unique mapping between domains.
>
>
> > “The authors argue that CycleGAN suffers from some drawback. Why do not the authors compare with CycleGAN in this paper?”
>
> Thank you for the observation that we could use better external baselines to compare against for domain transfer. We have added comparisons to pix2pix and CycleGAN for MNIST <-> Fashion MNIST. We find lower transfer accuracies and image quality (which we calculate with Frechet Inception Distance), which can be seen in Table 2 and Appendix C. The MNIST <-> MNIST scenario involved transferring between pretrained models with different initial conditions which is not directly comparable and has been omitted. In MNIST <-> SC09, the two domains were too distinct to provide any reasonable transfer with existing methods.
>
> As we mentioned in the paper, we also tried to train a CycleGAN between latent spaces, but weren’t unable to train the model at all, as the reconstruction loss was often trivially satisfied between models trained with the same Gaussian prior. This was an important finding for us, and gave us motivation to look at other methods for modeling transfer between latent spaces.
>
> > “the authors also need to compare with more state-of-the-art methods, such as StarGAN.”
>
> As mentioned above, thank you for pointing out the need for more baselines and we have now included comparisons to pix2pix and CycleGAN. We agree that StarGAN is an impressive model for multi-domain transfer, however, unlike the rest of the methods we compare, it requires additional target label information to be provided by the user at transfer time, which we feel makes CycleGAN a more natural comparison. Also, like CycleGAN, to the best of our knowledge these techniques still rely on structural similarities between domains and do not work as well for multi-modal transfer.
>
> > “Some implementation details are not clearly stated. ...how many labeled samples are used in Table 2?”
>
> As part of the paper revisions, we have done our best to make all the implementation details more explicit. For example, in the caption table 2, we discuss that we use all available labels (60k for MNIST<-> Fashion MNIST, 16k for MNIST <-> SC09). Table 3 then performs a comparison as the amount of data labels are reduced.

---

### Official Review · AnonReviewer2 · 2018-11-03
**A two-step solution for heterogeneous domain transfer (e.g., image-to-audio)**

**Rating:** 4
**Confidence:** 4

**Review:**

In this paper, the authors study an interesting problem, i.e., heterogeneous domain transfer such as knowledge transfer between an image domain and a speech/audio domain. In particular, the proposed solution contains two major steps: (i) pre-train each domain via VAE or GAN, and (ii) train a conditional VAE in semi-supervised manner in order to bridge two domains (see Section 2.2). Experiments on three public datasets (including three cross-domain settings) show the effectiveness of the proposed two-step solution.

Some Comments/suggestions:
(i) The technical novelty (considering the two-step solution) is limited though the studied problem is very interesting.

(ii) The authors are suggested to put the proposed solution in the context of transfer learning, which may better show the significance of this work. Currently, such a discussion and comparison is missing.

(iii) There are many grammar errors throughout the whole paper. The authors are suggested to significantly improve the linguistic quality.

(iv) A section of Conclusions is missing.

---

> ### Author Response · Authors · 2018-11-26
> **Response to AnonReviewer2**
>
> Thank you for your time and expertise in your review, we've addressed the key points below:
>
> > (i) The technical novelty (considering the two-step solution) is limited though the studied problem is very interesting.
>
> We would like to highlight that the problem this paper addresses (cross-modal domain transfer) is difficult and, to the best of our knowledge, relatively unexamined in the literature. We believe it is actually a desirable feature, and not a fault, that the proposed method is fairly straightforward and easy to implement.
>
> Similarly, we believe the two-step training actually has some important advantages over end-to-end training. First, this approach allows us to combine models that use dramatically different training procedures. We demonstrate that in this paper by transferring between a maximum-likelihood trained VAE and an adversarial-trained GAN. Second, for large generative models that take weeks to train, it would be infeasible to retrain the entire model for each new domain mapping. As a small example from this paper, training the bridging autoencoder from MNIST->SC09 takes ~1 hour on a single gpu, while retraining the SC09 WaveGAN takes ~4 days. We have also restricted ourselves to intuitive class-level mappings for the purpose of quantitative comparisons in this paper, but in a creative application it is likely each user would prefer their own unique mapping between domains.
>
> > “The authors are suggested to put the proposed solution in the context of transfer learning, which may better show the significance of this work. Currently, such a discussion and comparison is missing.”
>
> Thank you for the suggestion. Transfer learning does indeed share some surface similarities to the proposed work in that it uses pretrained networks. We would like to highlight, however, that transfer learning is actually quite distinct from domain transfer in that the pretrained networks are used as feature extractors for a new task, while in this work the pretrained networks are used for the same task on which they were trained (generating samples from a given distribution). Since no information is passing between the pretrained networks, the features learned in one domain are not informing the solution of generation in the other domain.
>
> > “There are many grammar errors throughout the whole paper. The authors are suggested to significantly improve the linguistic quality.”
> > “A section of Conclusions is missing.”
>
>
> We agree with your assessment and apologize for the rushed condition of the initial submission. You will hopefully find that the updated draft has been extensively revised and restructured to improve the clarity of the writing and the arguments, including adding a conclusion section.

---

### Official Review · AnonReviewer3 · 2018-11-07
**The technical part is weak**

**Rating:** 4
**Confidence:** 4

**Review:**

The authors demonstrate that it is possible to transfer across modalities (e.g., image-to-audio) by first abstracting the data with latent generative models and then learning transformations between latent spaces. We find that a simple variational autoencoder is able to learn a shared latent space to bridge between two generative models in an unsupervised fashion, and even between different types of models (e.g., variational autoencoder and a generative adversarial network). Some detailed comments are listed as follows,
1. The technical parts are weak since the authors use the existing method with to some extent evolution.

2 The proposed method can transfer the positive knowledge. However, for the transfer learning, one concerned and important issue is that some negative knowledge information can be also transferred. So how to avoid the negative transferring? Some necessary discussions about this should be given in the manuscript.

2 There are many grammar errors in the current manuscript. The authors are suggested to improve the English writing.

---

> ### Author Response · Authors · 2018-11-26
> **Response to AnonReviewer3**
>
> Thank you for your time and insight in your review. We’ve done our best to address your key points below.
>
> > The technical parts are weak since the authors use the existing method with to some extent evolution.
>
> We would like to highlight that the problem this paper addresses (cross-modal domain transfer) is difficult and, to the best of our knowledge, relatively unexamined in the literature. We believe it is actually a desirable feature, and not a fault, that the proposed method is fairly straightforward and easy to implement. From a technical standpoint, the main contribution is not a single new model with which to perform domain transfer, but showing it is possible to “glue together” the plethora of existing (and yet to be invented) models with small, simple, and efficient bridging models. While we have limited ourselves to several easily quantifiable problems for this paper, nothing about the proposed methods is limited to these models or datasets.
>
> > The proposed method can transfer the positive knowledge. However, for the transfer learning, one concerned and important issue is that some negative knowledge information can be also transferred. So how to avoid the negative transferring? Some necessary discussions about this should be given in the manuscript.
>
>
> Thank you for the suggestion. Transfer learning does indeed share some surface similarities to the proposed work in that it uses pretrained networks. We would like to highlight, however, that transfer learning is actually quite distinct from domain transfer in that the pretrained networks are used as feature extractors for a new task, while in this work the pretrained networks are used for the same task on which they were trained (generating samples from a given distribution). Since no information is passing between the pretrained networks, the features learned in one domain are not informing the solution of generation in the other domain.
>
> > There are many grammar errors in the current manuscript. The authors are suggested to improve the English writing.
>
>
> We agree with your assessment and apologize for the rushed condition of the initial submission. You will hopefully find that the updated draft has been extensively revised and restructured to improve the clarity of the writing and the arguments.

---

### Author Response · Authors · 2018-10-01
**A link to audio samples**

There was a mistake in the original that it didn't include a link to the audio samples. They are anonymously available  here:
https://drive.google.com/drive/u/8/folders/12u6fKvg0St6gjQ_c2bThX9B2KRJb7Cvk
Apologies.

---

### Meta-Review · Area_Chair1 · 2018-12-15
**Interesting problem, but work does not feel complete**

**Confidence:** 4
**Recommendation:** Reject

**Metareview:**

This paper studies the problem of heterogeneous domain transfer, for example across different data modalities.

The comments of the reviewers are overlapping to a great extent. On the  one hand, the reviewers and AC agree that the problem considered is very interesting and deserves more attention.

On the other hand, the reviewers have raised concerns about the amount of novelty contained in this manuscript, as well as convincingness of results. The AC understands the authors’ argument that a simple method can be a feature and not a flaw, however this work still does not feel complete. Even within a relatively simple framework, it would be desirable to examine the problem from multiple angles and "disentangle" the effects of the different hypotheses – for example the reviewers have drawn attention to end-to-end training and comparison with other baselines. The points raised above, together with improving the manuscript (as commented by reviewers) would make this work more complete.